# The Protective Effect of Cynara Cardunculus Extract in Diet-Induced NAFLD: Involvement of OCTN1 and OCTN2 Transporter Subfamily

**DOI:** 10.3390/nu12051435

**Published:** 2020-05-15

**Authors:** Francesca Oppedisano, Carolina Muscoli, Vincenzo Musolino, Cristina Carresi, Roberta Macrì, Caterina Giancotta, Francesca Bosco, Jessica Maiuolo, Federica Scarano, Sara Paone, Saverio Nucera, Maria Caterina Zito, Miriam Scicchitano, Stefano Ruga, Monica Ragusa, Ernesto Palma, Annamaria Tavernese, Rocco Mollace, Ezio Bombardelli, Vincenzo Mollace

**Affiliations:** 1Institute of Research for Food Safety & Health, Department of Health Sciences, University “Magna Graecia” of Catanzaro, 88100 Catanzaro, Italy; oppedisanof@libero.it (F.O.); muscoli@unicz.it (C.M.); xabaras3@hotmail.com (V.M.); carresi@unicz.it (C.C.); robertamacri85@gmail.com (R.M.); ketty.giancotta@gmail.com (C.G.); boscofrancesca.bf@libero.it (F.B.); jessicamaiuolo@virgilio.it (J.M.); federicascar87@gmail.com (F.S.); sara.paone06@gmail.com (S.P.); saverio.nucera@hotmail.it (S.N.); mariacaterina.zito@libero.it (M.C.Z.); miriam.scicchitano@hotmail.it (M.S.); rugast1@gmail.com (S.R.); mragusa@unicz.it (M.R.); palma@unicz.it (E.P.); an.tavernese@gmail.com (A.T.); rocco.mollace@gmail.com (R.M.); ezio.bombardelli@plantexresearch.it (E.B.); 2San Raffaele IRCCS, 00199 Rome, Italy

**Keywords:** insulin-resistance, hyperlipidemia, Non-Alcoholic Fatty Liver Disease (NAFLD), OCTN1, OCTN2

## Abstract

Hyperlipidemia and insulin-resistance are often associated with Non-Alcoholic Fatty Liver Disease (NAFLD) thereby representing a true issue worldwide due to increased risk of developing cardiovascular and systemic disorders. Although clear evidence suggests that circulating fatty acids contribute to pathophysiological mechanisms underlying NAFLD and hyperlipidemia, further studies are required to better identify potential beneficial approaches for counteracting such a disease. Recently, several artichoke extracts have been used for both reducing hyperlipidemia, insulin-resistance and NAFLD, though the mechanism is unclear. Here we used a wild type of Cynara Cardunculus extract (CyC), rich in sesquiterpens and antioxidant active ingredients, in rats fed a High Fat Diet (HFD) compared to a Normal Fat Diet (NFD). In particular, in rats fed HFD for four consecutive weeks, we found a significant increase of serum cholesterol, triglyceride and serum glucose. This effect was accompanied by increased body weight and by histopathological features of liver steatosis. The alterations of metabolic parameters found in HFDs were antagonised dose-dependently by daily oral supplementation of rats with CyC 10 and 20 mg/kg over four weeks, an effect associated to significant improvement of liver steatosis. The effect of CyC (20 mg/kg) was also associated to enhanced expression of both OCTN1 and OCTN2 carnitine-linked transporters. Thus, present data suggest a contribution of carnitine system in the protective effect of CyC in diet-induced hyperlipidemia, insulin-resistance and NAFLD.

## 1. Introduction

Hyperlipidemia, which includes hypercholesterolemia either associated or not associated with increased serum triglyceride levels, is often associated with Non-Alcoholic Fatty Liver Disease (NAFLD) [1,2,3]. On the other hand, NAFLD is closely associated with insulin-resistance and extra-hepatic disorders that involve the cardiovascular system, adipose tissue, intestines and muscles. For this reason, NAFLD may be defined as a systemic metabolic imbalance that increases cardio-metabolic risk [4,5,6,7].

Clear evidence exists that NAFLD is characterized by liver steatosis that is due to accumulation of fat in more than 5% of hepatocytes [7]. However, the pathophysiological mechanisms leading to accumulation of fat in the liver are still unknown and the development of novel therapeutic resources for approaching NAFLD still represents an unmet need. 

Clear evidence exists that imbalanced regulation of fat traffic in the liver is crucial for developing the disease and that elevated levels of circulating free fatty acids (FFAs) are associated with increased FFA transport proteins in the liver [8]. Moreover, a cumulative sequence of events contribute to the inability of hepatocytes to regulate the balance between FFAs inflow, hepatic de novo lipogenesis (DNL) and fatty acid efflux from liver tissue [9]. In particular, evidence has been accumulated that, in NAFLD, mitochondrial functionality is impaired mainly due to the increased mitochondrial oxidation deriving from greater FFA availability [10,11,12,13]. Furthermore, in NAFLD the oxidation of peroxisomes and microsomes occurring at the level of the endoplasmic reticulum (ER) is stimulated by a greater availability of hepatic FFA, the latter being re-esterified in triglycerides and assembled in VLDL (very low-density lipoproteins) [14,15,16,17,18,19]. 

The imbalanced modulation of FFAs in the liver of patients undergoing hyperlipidemia associated to NAFLD is accompanied by both regional as well as systemic inflammation. This is expressed by an increased number and activation of hepatic macrophages, enhanced signaling and local production of inflammatory cytokines and chemokines, which, at the late stages, leads to Non-Alcoholic Steato-Hepatitis (NASH), fibrosis and hepatocarcinoma [20,21,22,23,24]. 

Although evidence suggests that a clear relationship exists between elevated FFA circulating levels and the development of NAFLD, other mechanisms have recently been studied in order to better assess molecular mechanisms involved in fat accumulation in liver tissue and to identify novel therapeutic approaches for combating NAFLD and its consequences.

Recent data show that the modulation of the membrane transport system is crucial for the maintenance of homeostasis in many tissues including the liver and kidney, particularly under hyperlipidemic conditions [25,26]. In fact, these proteins are responsible for the uptake, elimination and intracellular transport of metabolites and nutrients [27,28]. Furthermore, membrane transporters are capable of interacting with xenobiotics, molecules with structures similar to substrates or alternative binding sites, thereby forming, for example, covalent bonds with cysteine residues [29]. Some of these transporters belong to the SLC22 family, which includes the OCT, OAT and OCTN subfamilies, being all characterized by broad substrate specificity [25]. In particular, OCTN subfamily (OCTN1 and OCTN2) seems to play a consistent role in the maintenance of general cellular homeostasis as they catalyze the carnitine transport across the membranes, thereby contributing to regulating carnitine levels in mammal cells [25,26,27,28,29,30]. Numerous studies have shown that OCTN1 and OCTN2 can be overexpressed or downregulated under different pathological conditions such as primary carnitine deficiency, diabetes, inflammatory bowel disease, neurological disorders, and cancer [25,26,27,28,29,30]. Furthermore, changes in carnitine metabolism have been reported under conditions of diabetes mellitus and obesity [25]. Therefore, it is likely that synthetic or natural compounds administered in these pathological conditions may have positive effects by modulating such transport mechanisms, leading to a beneficial effect in NAFLD. 

Cynara Cardunculus represents a solid component of a traditional Mediterranean diet [31], which displays potential lipid lowering and hepato-protective properties [32]. In particular, phytochemical studies revealed that Cynara Cardunculus Extract (CyC) is rich in antioxidants such as caffeic acid derivatives, (e.g., mono-caffeoylquinic acid and dicaffeoylquinic acid such as cynarin and chlorogenic acid), flavonoids (including the glycosides luteolin-7-beta-rutinoside, luteolin-7-beta-glucoside, and luteolin-4-beta-D-glucoside), and sesquiterpenes such as 5%–10% cynaropicrin [32,33]. The mechanism of action of CyC needs to be better clarified. However, it has been recently shown that luteolin, one of the CyC components, leads to a hypolipemic effect via inhibition of hydroxy-methyl-glutaryl-coenzyme A reductase, liver sterol regulatory element-binding proteins, and acetyl-CoA C-acetyltransferase [34], thereby leading to increased fecal excretion of sterols [35]. Furthermore, CyC synergizes with other lipid-lowering and hepatoprotective nutraceuticals, such as bergamot polyphenolic fraction [36,37], and could represent a consistent natural resource for combating the occurrence of combined hyperlipidemia and NAFLD. 

The present experiments have been performed to assess the potential beneficial effect of CyC on insulin-resistance, hyperlipidemia and NAFLD in rats fed a hyperlipidemic diet. Moreover, the potential contribution of modulation of OCTN 1 and OCTN2 membrane transporter subfamily in this experimental setting has also been investigated. 

## 2. Materials and Methods

### 2.1. CyC Preparation

Leaves from Cynara Cardunculus wild type were collected manually from spontaneous cultures in the geographical area located close to the Jonic sea in the Calabrian Region of Italy. Briefly, leaves from Cynara Cardunculus were crushed and washed three times at high pressure water flow. Then the juice obtained was filtrated in columns and then eluted with KOH solution. The fluid from columns was then concentrated and desiccated to obtain CyC. This was also enriched with active ingredients obtained via hydro-alcoholic extraction of Cynara Cardunculus leaves to obtain the final concentration. Analysis of CyC via Orbitrap (Thermo Scientific, Milan) revealed 10% cynaropicrin, 12% of total flavonoids (expressed as luteolin-7-O- glucoside) and 6% caffeoylquinic acid (expressed as Chlorogenic acid). 

### 2.2. Animals

Male Sprague-Dawley rats (270–290 g, Charles River, Milan, Italy) were used throughout the study. All animals were housed and cared in accordance with Italian National Health Ministry Guidelines on Laboratory Animal Welfare following the Italian regulations for the protection of animals used for experimental and other scientific purposes (D.L. 26/2014), and with European Economic Community regulations (2010/63/UE). The numbers of animals used were the minimum necessary to achieve statistical significance at *p* < 0.05. Rats were housed two per cage and maintained under identical conditions of temperature (21 ± 1 °C) and humidity (60% ± 5%), with a 12-hour light/12-hour dark cycle and allowed food ad libitum. All experiments took place during the light period in a quiet room.

Assessment of feeding behavior throughout the study was carried out according to European Association for the Study of the Liver (EASL) Guidelines., [38] revealing that no changes occurred among different treatment groups. In addition, no changes in body temperature and ketone bodies throughout the study were observed.

High-fat diet TD.88137 Total Fat (21% by weight; 42% kcal from fat) was purchased from Harlan Laboratories, Rossdorf, Germany; CyC (Cynara Cardunculus leaf extract) was kindly provided by H&AD (Herbal and Antioxidants Derivatives srl, Bianco, Italy). Test products were dissolved in water. 

### 2.3. Study Design

The study design is illustrated in the Figure 1.

After an adaptation period of one week, rats were allocated into one of the following experimental groups: 

Control group (*n* = 6), fed a Normal Fat Diet (NFD) for four weeks; 

HFD group (*n* = 6), this group received a High-Fat Diet (HFD) for four weeks; 

NFD receiving 10 mg/Kg of CyC for 4 consecutive weeks (*n* = 6);

NFD receiving 20 mg/Kg of CyC for 4 consecutive weeks (*n* = 6);

HFD receiving 10 mg/Kg of CyC for 4 consecutive weeks (*n* = 6); 

HFD receiving 10 mg/Kg of CyC for 4 consecutive weeks (*n* = 6);

All treatments were given via gastric gavage once daily over a period of four weeks. Body weight was measured before starting treatment and at the end of the feeding period.

### 2.4. Blood Biochemical Analysis

On day 1 and after 4 weeks, for each group a small blood volume was collected by simply puncturing the tail vein with a small gauge needle in order to determine serum glucose levels, total cholesterol and triglyceride levels. All determinations were found in animals fasted overnight. Measurements were performed by means of enzymatic assay commercial kits (Multicare, Milan, Italy) according to the manufacturer protocol.

A StetoTest, which included multiple biomarkers of NAFLD such as serum a2-macroglobulin, apolipoprotein A1, haptoglobin, total bilirubin and gamma-glutamyltranspeptidase, alanine-aminotransferase (ALT) plus body mass index, serum cholesterol, triglycerides and glucose, was performed in all groups of animals. [39].

The index of the oxidative stress was evaluated by assessing serum lipid peroxidation product malondialdehyde (MDA) using a lipid peroxidase assay kit (Sigma-Aldrich, Saint Louis, MO, USA) according to the manufacturer’s protocol. Briefly, the serum sample was first treated with trichloroacetic acid (TCA) for protein precipitation and then treated with thiobarbituric acid. The mixture was heated for 10 min in a boiling water bath. One molecule of MDA reacts with two molecules of thiobarbituric acid. The resulting chromogen was centrifuged and the intensity of colour developed in supernatant was measured colourimetrically at 530 nm.

### 2.5. Morphological Analysis of Liver

After completion of the treatments, the rats of each group were randomly divided in two experimental categories: histological analysis and biochemical investigations. All animals were anesthetized with an intramuscular injection of 100 mg/kg ketamine and 5 mg/kg xylazine, and for biochemical assay, the livers were removed, immersed in liquid nitrogen and subsequently stored at −80 °C. For histological analysis, livers were fixed by transcardial perfusion at 120 mmHg with 100 mL of Phosphate-Buffered Saline (pH 7.2), followed by 150 mL of 4% paraformaldehyde (pH 7.2). Subsequently, tissue samples were fixed by immersion in 10% buffered formalin for about 48 h. The specimens were processed using automatic tissue processor for histology (VTP 300-Bio Optica) and embedded in paraffin. Tissue sections of 5 µm thickness were cut by means of microtome (Microtome pfm Rotary 3003-Bio Optica), placed on slides and stained with Haematoxylin & Eosin stains (H&E) for further histological examination. H&E stained sections of the liver were examined with the light microscope using a magnification of ×100 and ×400 and transformed into digital images using a camera and specific software (Olympus).

### 2.6. Western Blotting Analysis

Collected tissues were immersed in a lysis buffer containing 250 mM Sucrose, 10 mM Tris/HCl pH 7.8, 1 mM EDTA and the protease and phosphatase inhibitor. Each tissue was homogenized at 4 °C using a 2 mL Potter–Elvehjem Tissue Grinders with a PTFE Pestle, and then, after 15 min at 4 °C, centrifuged at 12,000× *g* for 15 min at 4 °C. Supernatants were subjected to protein analysis using a commercially available protein assay kit (Bio-Rad, Watford, Hertfordshire, WD17 1ET, UK). Protein concentration was determined by the Lowry procedure using bovine serum albumin as standard. Liver lysates from each treatment (20 μg protein/lane) were resolved in 10% or 12% Sodium Dodecyl Sulfate–polyacrylamide gel (SDS-PAGE) mini-gels and electrophoretically transferred to nitrocellulose membranes. Residual binding sites on the membrane were blocked by incubation with 5% non-fat dried milk, (Sigma-Aldrich, St. Louis, MO, USA) in a buffer solution composed of NaCl 140 mM, 20 mM Tris–HCl, Tween 20 0.05% pH 7.6, for 60 min at room temperature followed by an overnight incubation at 4 °C with a rabbit polyclonal anti-OCTN2 (1:2,000; Sigma-Aldrich, St. Louis, MO 63178, USA) and a rabbit polyclonal anti-OCTN1 (1:1,000; Sigma-Aldrich, St. Louis, MO 63178, USA). The blots were then washed three times with washing buffer and were incubated with an anti-rabbit IgG-coupled horseradish peroxidise antibody (Thermo Fisher Scientific, Waltham, MA 02,451 USA) for 1 h at room temperature. A mouse monoclonal β-actin antibody (2 h at room temperature, dilution 1:1000; Sigma-Aldrich, St. Louis, MO 63178, USA) was used for the loading control. After washes, proteins were visualized by enhanced chemiluminescence (ECL; GE Healthcare, Boston, MA 02210, USA) according to the manufacturer’s instructions. The amount of reaction and quantitative evaluations was estimated by the UVITEC Chemiluminescence Documentation System. No difference for β-actin was detected among the lanes. 

### 2.7. Statistical Analysis

Data were analyzed with GraphPad PRISM 7.0 (GraphPad Software, Inc., La Jolla, CA, USA). Results are shown as mean ± SEM. Normally distributed data were analysed by one-way ANOVA followed by Tukey’s test, while data without normal distribution were analyzed using Kruskal–Wallis analysis of variance and subsequent Dunn’s tests. A *p*-value < 0.05 was considered statistically significant.

## 3. Results

### 3.1. The Effect of CyC on Serum Glucose, Cholesterol, Triglyceride Levels and Body Weight in NFD and HFD Rats

Serum glucose, total cholesterol, triglyceride, SteatoTest and MDA levels at baseline were similar in all four groups of rats used throughout the study. In rats fed an HFD over a period of four consecutive weeks, an increase of serum glucose, total cholesterol, triglyceride levels, SteatoTest, and MDA was found compared to rats receiving standard diet (NFD) (Table 1). This effect was counteracted dose-dependently by supplementation of rats with CyC. In particular, in animals fed a HFD and supplemented with 10 mg/kg of CyC in a single daily administration via gastric gavage, serum glucose, total cholesterol, triglyceride, SteatoTest and MDA levels were significantly lower than the concentrations found in rats receiving only HFDs. Moreover, a further increase of hypoglycemic and anti-lipemic effect of CyC was seen when a dose of 20 mg/kg was used (Table 1). No effect was found in rats receiving a standard diet (NFD) and supplementation with CyC10 and 20 mg/kg. The effect of CyC on serum glucose, triglycerides, cholesterol, SteatoTest and MDA found in rats fed a HFD was associated with a significant reduction in body weight. In fact, feeding rats over a four weeks period with HFD produced a significant body weight increase (Figure 2). This effect was counteracted significantly by oral supplementation of animals with CyC 10 and 20 mg, dose-dependently (Figure 2).

### 3.2. CyC Supplementation Improves HFD-Related Liver Steatosis in Rats

After four weeks of study, histological analysis did not show liver damage in rats receiving normal diet (NFD) (Figure 3A). In contrast, histopathological study of liver parenchyma showed that feeding animals with HFDs induced hepatic steatosis. In particular, HFD-fed rats displayed prominent hepatic steatosis at the end of four weeks of treatments (Figure 3B). In particular, cytoplasmic alterations found in hepatocytes of rats fed a HFD revealed the occurrence of microvesicular steatosis; the cytoplasm had been replaced by bubbles of fat that did not displace the nucleus, while signs of macrovesicular steatosis had not been found. In Figure 3C,D it is shown that CyC (10 mg/kg or 20 mg/kg) supplementation was able to prevent steatosis found in HFD-fed rats at the end of four weeks of treatments.

### 3.3. CyC Supplementation Counteracts OCTN1 and OCTN2 Transporter Reduction in Liver Tissue of Rats Fed HFD

OCTN1 and OCTN2 have been found to be expressed in liver tissue of rats under basal conditions and remained unchanged after feeding rats with NFD over four weeks of treatment (Figure 4). In rats fed a HFD, a significant reduction of both OCTN1 and OCTN 2 expression was found, with the reduction of both transporters being nearly equivalent. Supplementation of rats with CyC restored OCTN1 and OCTN2 expression in liver tissues after four weeks. However, only the dose of 20 mg/kg was found to be effective in restoring completely and significantly OCTN1 and OCTN2 expression in rats fed a HFD (Figure 4). No change was found in OCTN1 and OCTN2 expression in rats fed NFD after receiving supplementation with CyC (10 and 20 mg/Kg daily) for four consecutive weeks.

## 4. Discussion

The present data show that CyC, a sesquiterpene and antioxidant-rich artichoke leaf extract, antagonized metabolic imbalance and overweight in rats fed a HFD. This occurs after four weeks supplementation, an effect which occurs dose-dependently and involves both hyperglycemia and elevated serum levels of total cholesterol and triglycerides. In addition, daily supplementation with CyC antagonized liver steatosis in rats fed a HFD, suggesting that active ingredients found in the herbal extract from wild types of artichoke produce simultaneous antagonistic effects on both diet-induced metabolic imbalance and liver injury. 

This is in accordance with previous data suggesting that artichoke derivatives possess metabolic regulatory properties and that an overall improvement of NAFLD condition may be detected when such a nutraceutical supplementation occurs both in experimental models of liver dysfunction and in patients [31,32].

In particular, evidence exists that Cynara extracts produce an improvement of lipid metabolism in liver cells [31,32,40]. Several mechanisms have been implicated in hypolipemic effects of nutraceutical supplementation with Cynara derivatives based on biomolecular studies carried out both in vitro and in vivo [31,32,41,42]. In fact, the use of Cynara leads to more efficient uptake and processing of chylomicron remnants deriving from diets of hepatocytes. On the other hand, hepatic apolipoprotein biosynthesis is stimulated by artichoke extracts with the consequence of an improved uptake of LDLs and HDLs [43,44,45]. This also leads to stimulation of hepatic HDL synthesis driven by increased formation of apolipoproteins A1 and A2. Finally, Cynara extracts seem to enhance HTGL and Lecithin-Cholesterol Acyl Transferase LCAT biosynthesis in hepatocytes, thereby explaining both hypolipemic and hepato-protective activities we found with CyC supplementation in rats fed a HFD.

Alongside hypolipidemic properties of CyC, hypoglycemic effects seen in rats fed a HFD are also supported by previous evidence in vitro and in vivo. In particular, it has been shown that chlorogenic acid (a crucial component of CyC extract used throughout the study) produces inhibition of glucose intestinal absorption, an effect accompanied by marked inhibition of glucose-6-phosphate-translocase [46,47,48], a key enzyme in de novo glucose biosynthesis. This is confirmed by studies revealing that caffeoylquinic acids inhibit glucosidases, thereby explaining the reduction of serum glucose levels found in rats fed a HFD and receiving CyC supplementation [46,47].

These effects have been confirmed in patients via double-blind, placebo-controlled clinical trials carried out in subjects with hyperglycemia and that are overweight in which supplementation with Cynara extracts produced consistent benefits in counteracting insulin-resistance [46,47].

The beneficial effect of Cynara derivatives in NAFLD was confirmed by our data in rats fed a HFD. This seems to be related to the high concentration of cynaropicrin in CyC extract. Indeed, evidence exists that cynarin and caffeoylquinic acids lead to liver protection in models of liver injury produced by hepatotoxic agents such as carbon tetrachloride, an effect accompanied by reduced oxidative stress and inflammation, as indicated by lower concentrations of liver malondialdehyde and serum transaminases in rats treated with Cynara derivatives [48,49,50]. Moreover, Cynara extract leads to regeneration on injured liver cells, an effect confirmed when using luteolin and other polyphenols [48]. These active ingredients have also been found to possess cholaretic properties, which, at least in part, should contribute to the hepatoprotective effect of Cynara extract found in both in vitro and in vivo settings [50,51,52,53,54].

Besides, the mechanism which associates lipid lowering properties of Cynara derivatives and NAFLD still remains unclear.

Our data show for the first time that CyC restored, at higher dose, OCTN1 and OCTN2 carnitine-linked transporter expression in liver tissue of rats fed a HFD. 

It is well known that the carnitine system, including transport mechanisms involving the subfamily of transporters OCTN1 and OCTN2, contribute in the modulation of fatty acid and carbohydrate metabolism [25]. In humans, 75% of total carnitine derives from the diet and only 25% is synthesized in kidney, liver and brain [25,26,27,28]. Alterations of carnitine metabolism have been recently linked also to diabetes mellitus and obesity [29,30]. The main function of carnitine is to shuttle acyl and acetyl groups from the mitochondrial matrix to hepatic cytosol and other tissues. Thus, carnitine would allow the export of fatty acids from tissues in the form of acylcarnitines, resulting in a reduction in lipid-induced insulin resistance and a consequent increase in glucose utilization [25]. This occurs in HFDs in which microvesicular steatosis is associated with reduced expression of hepatic OCTN1 and OCTN2. Indeed, it is likely that the reduced expression of OCTN transporters induced by HFD leads to a deficit of carnitine, with a consequent reduction in the transfer of long chain fatty acids from the cytosol to the mitochondria and lack of oxidation. This, in turn, is accompanied by accumulation of lipids in the liver thereby leading to hepatic steatosis. These effects are reversed when the higher dose of 20 mg/kg of CyC was used for supplementing HFD rats, with this effect needing to be better clarified.

It is known that errors in nuclear receptor signaling, including peroxisome proliferator-activated receptor alpha (PPARα), are involved in the pathogenesis of fatty liver disease [25,55]. Furthermore, it has been shown that PPARα upregulates OCTN transporter subfamily expression, an effect driven by natural active ingredients as the ones found in CyC [56]. Thus, it is likely that CyC, via activation of PPARα, counteracts the effect of HFDs in OCTN transporter expression and carnitine availability, thereby contributing in hepato-protective activity of CyC. 

Finally, our data also show that the CyC extract produced weight loss in HFDs. This effect occurred without consistent changes in food intake and in energy expenditure, compared to rats treated with NFD, thus suggesting that other mechanisms may contribute to this response. The effect of artichoke extract (both Cynara Scolimus and Cynara Cardunculus) on weight loss has widely been studied in the last few years. This activity, found both in animals and in patients with metabolic syndrome, has been explained on the basis of different mechanisms. These include the lowering of inflammation as evidenced by the reduced plasma interleukin (IL)-6, IL-1β, and plasminogen activator inhibitor-1 levels, mostly due to the effect of luteolin [57]. On the other hand, other active ingredients of artichoke extract, such as inulin, have been found to produce body weight reduction alongside hypoglycemic effects in obese animals [58]. Further experiments are required in order to better clarify this additional effect of artichoke derivatives.

## 5. Conclusions

In conclusion, our data show that CyC, a Cynara Cardunculus wild type leaf extract, counteracts hyperglycemia and elevated serum levels of total cholesterol and triglycerides in rats fed a HFD over a four week period. This effect is associated with significant inhibition of liver steatosis seen in rats fed a HFD, thus suggesting that active ingredients found in the herbal extract from the wild type of artichoke produce simultaneous antagonistic effects on both diet-induced metabolic imbalance and liver injury. The beneficial effect of CyC seems to involve the carnitine transport system as OCTN1 and OCTN2 transporters linked to carnitine in the liver tissue of rats fed a HFD are enhanced by artichoke extract. Overall, these results shed new light in the therapeutic strategy for counteracting metabolic disorders associated to NAFLD. 

## Figures and Tables

**Figure 1 nutrients-12-01435-f001:**
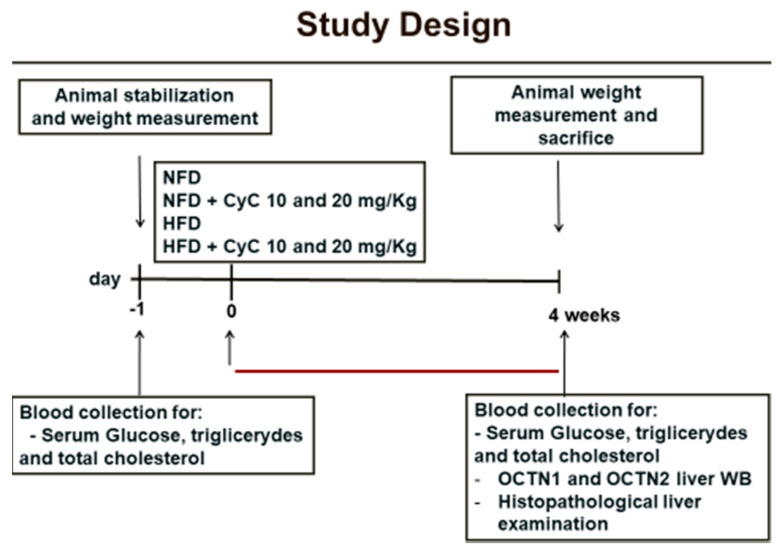
Study design.

**Figure 2 nutrients-12-01435-f002:**
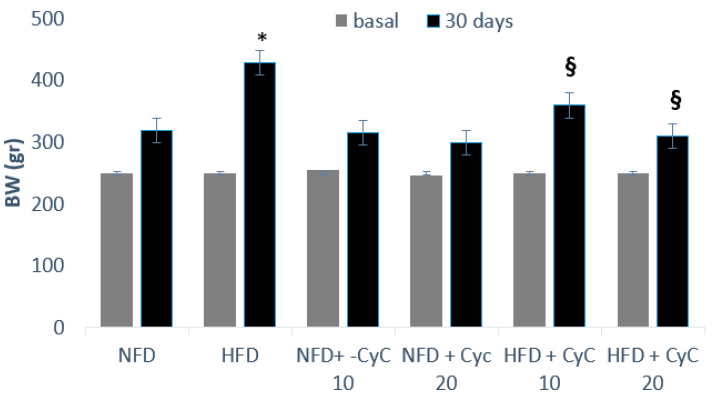
The effect of CyC (10 and 20 mg/kg daily given orally over a period of four weeks) on body weight in Normal Fat Diet (NFD) and High Fat Diet (HFD) groups. Data are expressed as mean ± SE.* *P* < 0.05 HFD vs. NFD. ^§^
*P* < 0.05 HFD + CyC vs. HFD.

**Figure 3 nutrients-12-01435-f003:**
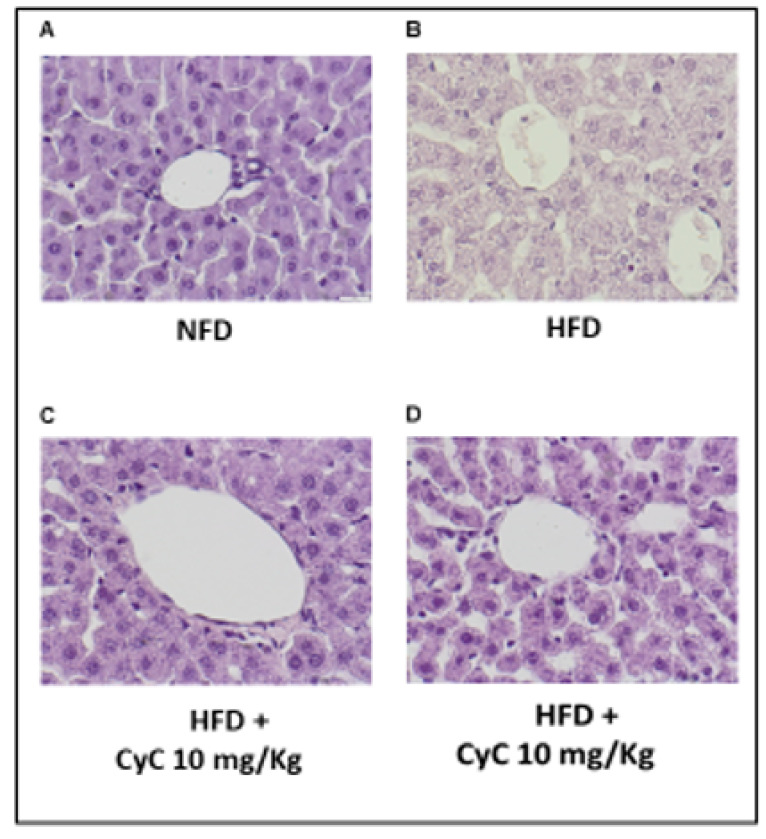
The effect of CyC (10 and 20 mg/kg given daily and orally over a period of four weeks) on histopathological features of liver steatosis in NFD and HFD. Histopathological sections have been stained with hematoxylin-eosin.

**Figure 4 nutrients-12-01435-f004:**
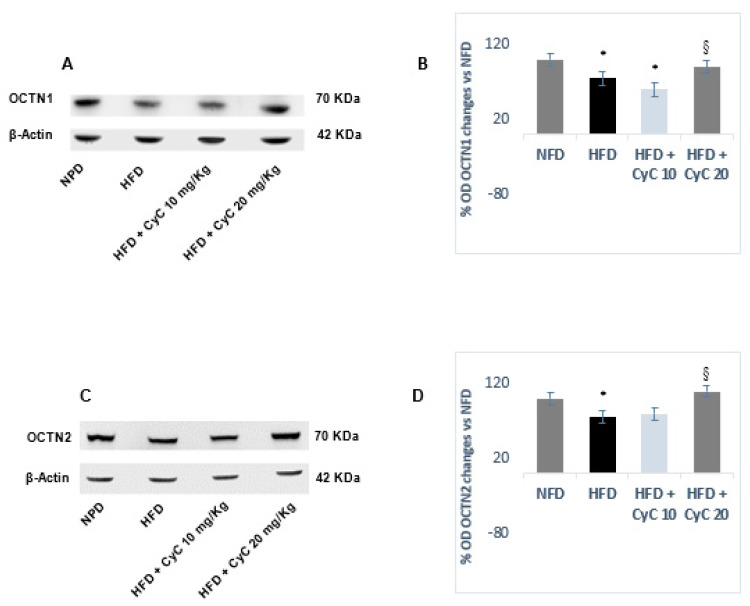
The effect of CyC (10 and 20 mg/kg daily given orally over a period of four weeks) on OCTN1 (**A**,**B**) and OCTN2 (**C**,**D**) carnitine-linked transporters in NFD and HFD. Representative Western blotting analysis are expressed in subgraph A and C. Optical Density for each blot is expressed in Subgraph B and D). Data are expressed as mean ± SE.* *P* < 0.05 HFd vs. NFD. ^§^
*P* < 0.05 HFD + CyC vs. HFD.

**Table 1 nutrients-12-01435-t001:** The effect of Cynara Cardunculus extract (CyC) (10 and 20 mg/kg daily given orally over a period of four weeks) on serum glucose (mg/dL), total cholesterol (mg/dL), triclycerides (mg/dL), SteatoTest (ST score) and MDA (nm/μl) in NFD and HFD. Data are expressed as mean ± SE.

Parametres		NFD (*n* = 6)	HFD (*n* = 6)	NFD + CyC (10 mg/Kg) (*n* = 6)	NFD + CyC (20 mg/Kg) (*n* = 6)	HFD + CyC (10 mg/Kg) (*n* = 6)	HFD + CyC (20 mg/Kg) (*n* = 6)
**Serum Glucose**							
	Basal	62 ± 2	65 ± 3	68 ± 3	64 ± 4	66 ± 5	66 ± 5
	4 weeks	2 ± 1	22 ± 4 *	3 ± 1	2 ± 0,5	15 ± 5 ^§^	4 ± 2 ^§^
**Total Cholesterol**							
	Basal	135 ± 5	138 ± 4	138 ± 5	141 ± 6	136 ± 6	138 ± 5
	4 weeks	4 ± 1	38 ± 3 *	2 ± 1	2 ± 0.5	18 ± 3 ^§^	6 ± 2 ^§^
**Triglycerides**							
	Basal	145 ± 5	143 ± 4	146 ± 4	144 ± 4	144 ± 5	147 ± 4
	4 weeks	4 ± 1	48 ± 3 *	4 ± 1	2 ± 0,5	27 ± 4 ^§^	8 ± 2 ^§^
**SteatoTest**							
	Basal	0.40 ± 0.05	0.42 ± 0.04	0.41 ± 0.04	0.43 ± 0.03	0.41 ± 0.05	0.40 ± 0.04
	4 weeks	0.12 ± 0.02	0.42 ± 0.03 *	0.14 ± 0.021	0.16 ± 0.03	0.22 ± 0.04 ^§^	0.36 ± 0.02 ^§^
**MDA**							
	Basal	1.1 ± 0.05	1.2 ± 0.04	1 ± 0.04	1.2 ± 0.03	1.2 ± 0.05	1 ± 0.05
	4 weeks	0.1 ± 0.05	0.7 ± 0.03 *	0.2 ± 0.05	0.1 ± 0.05	0.4 ± 0.04 ^§^	0.6 ± 0.03 ^§^

* *P* < 0.05 HFD vs. NFD. § *P* < 0.05 HFD + CyC vs. HFD.

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
