# Peer review of "The Protective Effect of Cynara Cardunculus Extract in Diet-Induced NAFLD: Involvement of OCTN1 and OCTN2 Transporter Subfamily"

_nutrients, 2020, doi:10.3390/nu12051435_

Round 1

Reviewer 1 Report

The authors investigated the positive effect of Cynara Cardunculus extract (CyC) for non-alcoholic fatty liver disease (NAFLD). They found that the agent resulted in fatty liver improvement and the effect was partly based on OCTN1 and 2. While the contents of this study are interesting, there are several points that still need to be addressed.

Major

  1. The change in NAFLD pathogenesis by Cyc administration is not well defined. There included only the body weight, blood cholesterol levels, and simple liver specimen findings. The fat contents or fat metabolism related marker changes in the liver should be added.
  2. In the liver specimen evaluation, fatty liver is improved even in the low concentration of Cyc. However, the change in OCTN 1 and 2 were evident only in the high concentration of Cyc. This result indicates that OCTNs changes are not the effect of Cyc on the fatty liver changes. The authors should find another factors that show positive effects even in the lower concentration of Cyc.

Author Response

ANSWERS TO THE POINTS RAISED BY THE REFEREES

REVIEWER N. 1

  1. The change in NAFLD pathogenesis by Cyc administration is not well defined. There included only the body weight, blood cholesterol levels, and simple liver specimen findings. The fat contents or fat metabolism related marker changes in the liver should be added.

Thanks for the suggestion. The results of steato-test, a comprehensive biomarker for NAFLD approved  by the European Associations operating in this field (see Test analysis of European Association for the Study of the Liver (EASL); European Association for the Study of Diabetes (EASD); European Association for the Study of Obesity (EASO). EASL-EASD-EASO Clinical Practice Guidelines for the management of non-alcoholic fatty liver disease. J Hepatol 2016; 64:1388–1402.) have now been included in the revised manuscript.

  1. In the liver specimen evaluation, fatty liver is improved even in the low concentration of Cyc. However, the change in OCTN 1 and 2 were evident only in the high concentration of Cyc. This result indicates that OCTNs changes are not the effect of Cyc on the fatty liver changes. The authors should find another factors that show positive effects even in the lower concentration of Cyc.

Data on MDA for oxidative stress measurements, have now been added and appear in the final version of the manuscript. However, we would like to stress once again the concept that mechanisms other than oxidative stress and inflammation (which have been widely studied; see Musolino et al., 2020) such as OCTN transporter subfamily, may contribute in the pathogenesis of NAFLD.

Reviewer 2 Report

The ms. entitled „The protective effect of …” by Oppedisano et al. studies the effect of Cynara Cardunculus on hepatic steatosis in HFD-fed rats. The authors describe a dose-dependent anti-steatotic effect and an improvement of both dyslipidemia and glycemia by the extract. This was accompanied by a decreased body weight and restored expression of carnithin transporters.

The study is of principal interest. However, further data are required, to allow a clear conclusion.

  1. The anti-steatotic effect and the beneficial effects on plasma lipids and glucose can easily be explained by the body weight loss in response to the extract. What is the reason for the reduced body weight in response to the plant extract? The authors should provide data that address the effect of the extract on food intake and exclude a direct effect of the extract on food Aversion and sickness, as well as nutrient resorption.
  2. Is there evidence for increased energy expenditure (core body temperature)? Were ketone bodies altered as a marker for increased beta-oxidation?
  3. For how long the animals received the extract: according to Fig. 1 for weeks, but in the section 2.3. it was stated 4 days, if that is true: at the beginning or at the end of the HFD?
  4. Did the controls receive a vehicle? What was it?
  5. How long after the last dosing of the extract was the blood obtained? As it was most likely in the fed state, how can the authors exclude an acute effect of the extract on food intake that would impact the plasma parameters?

Author Response

ANSWERS TO THE POINTS RAISED BY THE REFEREES

REVIEWER N. 2

The study is of principal interest. However, further data are required, to allow a clear conclusion.

  1. The anti-steatotic effect and the beneficial effects on plasma lipids and glucose can easily be explained by the body weight loss in response to the extract. What is the reason for the reduced body weight in response to the plant extract? The authors should provide data that address the effect of the extract on food intake and exclude a direct effect of the extract on food Aversion and sickness, as well as nutrient resorption.

An assessment of feeding behaviour according to headlines summarized by Ellacot et al., (Assessment of feeding behavior in laboratory mice Cell. Mateb.2016, 4,12,10-17) has been made throughout the study with no changes found among animal groups. This is now dysplayed in the manuscript.

  1. Is there evidence for increased energy expenditure (core body temperature)? Were ketone bodies altered as a marker for increased beta-oxidation?

No changes were found in energy expnditure or ketone bodies in any of the groups entering the study. This now appears in the manuscript

  1. For how long the animals received the extract: according to Fig. 1 for weeks, but in the section 2.3. it was stated 4 days, if that is true: at the beginning or at the end of the HFD?

A mistake occurred in the section 2.3  on the time of treatment.The due changes have been included in the revised manuscript

  1. Did the controls receive a vehicle? What was it?

The test product was dissolved in water and this was used in all the treatment group including the control. This now is better specified

  1. How long after the last dosing of the extract was the blood obtained? As it was most likely in the fed state, how can the authors exclude an acute effect of the extract on food intake that would impact the plasma parameters?

All measurements were made via blood samples collected in animals which were fasted overnight. This is now described in the revised manuscript

Round 2

Reviewer 1 Report

The authors responded to the comments adequately.

Author Response

Thenks for your positive evaluation

Reviewer 2 Report

The changes made by the authora significantly improved the manuscript. However, the authors should state their hypothesis, why body weights were reduced in the treated animals, in view that food intake and energy expenditure were not changed.

Author Response

Reviewer:

The changes made by the author significantly improved the manuscript. However, the authors should state their hypothesis, why body weights were reduced in the treated animals, in view that food intake and energy expenditure were not changed

Authors

The effect of artichoke extract (both Cynara Scolimus and Cynara Cardunculus) on weight loss has widely been shown in the last few years. This activity, found both in animals and in patients with Metabolic syndrome, has been explained on the basis of different mechanisms. These include the lowering of inflammation as evidenced by the reduced plasma interleukin (IL)-6, IL-1β, and plasminogen activator inhibitor-1 levels., mostly due to the effect of luteolin (Kwon EY, Kim SY, Choi MS  Luteolin-Enriched Artichoke Leaf Extract Alleviates the Metabolic Syndrome in Mice with High-Fat Diet-Induced Obesity..Nutrients. 2018 Jul 27;10(8):979). On the other hand, other active ingredients of artichoke extract, such as inulin, have been found to produce body weight reduction alonside with hypoglycemic effect in obese animals (Shao T, Yu Q, Zhu T, Liu A, Gao X, Long X, Liu Z. Inulin from Jerusalem artichoke tubers alleviates hyperglycaemia in high-fat-diet-induced diabetes mice through the intestinal microflora improvement. Br J Nutr. 2020 Feb 14;123(3):308-318).

These additional informations have now been included and appear in the revised manuscript.